# Sustainable Development Planning: Master's Based on a Project-Based Learning Approach

**Adolfo Cazorla-Montero [1,†], Ignacio de los Ríos-Carmenado [1,*,†]  and Juan Ignacio Pasten [2,†]**

1    Planning and Sustainable Management of Rural-Local Development (GESPLAN), Universidad Politécnica de Madrid, Avda Puerta de Hierro 2, 28040 Madrid, Spain; adolfo.cazorla@upm.es
2    Facultad de Ingeniería Agraria, Universidad Católica Sedes Sapientiae, Lima 15302, Peru; ji.pasten@alumnos.upm.es
*    Correspondence: ignacio.delosrios@upm.es; Tel.: +34-610-877-699
†    These authors contributed equally to this work.

**Abstract:** The educational subject of Sustainable Development Planning in Europe is evolving due to the implementation of the Bologna Agreement across the European Higher Education Area (EHEA). This paper analyses a project-based learning strategy for training Sustainable Development Planning in postgraduate programs, in Spain (Universidad Politécnica de Madrid, UPM). This project-based learning strategy is applied to an International Postgraduate Program for Sustainable Rural Development—Erasmus Mundus, Master's of Science—with the participation of five European Union universities that formed the Agris Mundus Alliance for Sustainable Development. Using a mixed methods approach, the research examined the program's implementation through student and staff perceptions, from the technical, behavioral and contextual project management skills. The paper argues that the "Practical Learning platforms" used in the Master's demonstrate the correct approach of the learning strategy based on teaching–research linked to the professional sphere. The findings that were identified can be categorized as follows: (1) Perspective: holistic thinking and intellectual coherence, defining the contextual skills that must be navigated within and across the broader environment, (2) Practice: experiential learning by reconnecting to real-life situations, and (3) People: Personal and interpersonal skills required to succeed in sustainable projects, programs and portfolios. Reflections on the experience and main success factors in the learning strategy are discussed.

**Keywords:** rural development; Bologna Agreement; project management; sustainable development; sustainability competencies; transdisciplinary sustainability

## 1. Introduction

Planning and Sustainable Development concepts are deeply linked in both the public and private domains. They come together in what could be called "operability" and are academically intertwined because they share teaching and research topics. The relationship between plans, programs and projects is traditionally included in technical universities' curricula. The European dimension of planning for rural development is relevant for at least three reasons: the need for the rural planner to be aware of the different territorial planning systems that operate across the EU; the increase in cross-border planning policies and other transnational project planning for rural development; and the European level in the hierarchy of planning levels [1].

Currently, we are involved in a wide-reaching process of reflection and change aimed at promoting a qualitative leap in the educational models of universities in the European Union. This approach stems

from different agreements reached in the EU to establish the European Higher Education Area (EHEA) as the basis of a new knowledge-based economy that responds to the challenges of globalization.

However, the starting point of any strategy should be a clear idea of the university's end goal and purpose. Whatever model is adopted, one thing should remain unaltered in the university institution—the incessant search for truth in which teachers teach what they discover day to day, subjecting their own knowledge to permanent criticism, with a marked vocation for service to the society in which they are immersed [2]. From this relevant vocation of service to society in which the university is immersed, sustainable development models have been generated, linking research and teaching, especially from the so-called world-class universities [3,4].

The high level of qualifications, thinking about higher education is currently equivalent to thinking about creating world-class universities [4,5]. The prestige of the researchers and the effectiveness and transferability of knowledge, which they provide both the public and private sectors, demonstrate that research universities represent a new successful educational model that generates knowledge and interacts with society [4].

The internationalization of higher education and rise in international student mobility was accompanied by much reflection on the extent to which universities were equipping their students to be future world citizens or "world professionals" [6]. In this context, the universities began methodological changes with approaches focused on new competences and with the aim of improving knowledge and aptitudes under a personal, civic, social, or job-related perspective [7].

In this context, currently, we are involved in a wide-reaching process of reflection and change aimed at promoting a qualitative leap in the educational model of European Union universities as the basis of a new knowledge-based economy that responds to the challenges of globalization. The EHEA stresses that one of the measures required for achieving employability is developing transversal skills and competencies, including competency in communication and languages, the ability to handle information, the ability to solve problems and to work in teams, and the ability to lead social processes [8]. The concept of competency is an essential foundation in the professional world and is a key element of any educational model. Competence is an amplification of the concepts of ability and qualification resulting from a rapid technical evolution in the organization of work and planning activities. Most enterprises today demand competent professionals.

On the other hand, sustainability and sustainable planning are a priority theme in most governments' higher education agendas all over the world and transversal skills that are considered in world-class universities' curricula [4,5]. Sustainable planning and project concepts are deeply linked in both the public and private domains—coming together in what could be called "operability"—and are intertwined academically because they share teaching and research topics. The relationship between plans, programs, and projects is traditionally included in technical universities' curricula, and the future of sustainability planning is dependent upon the education of an interdisciplinary workforce with broad and holistic skills related to environmental, social, and economic stewardship [9].

It is fundamental to consider the aptitude and abilities that society demands of its future professionals—including helping the next generation of practitioners and sustainable planning leaders—when designing any educational strategies. At the basis of the education there must be a core understanding of sustainability planning and the development of skills to contribute to sustainability management issues.

Within this general framework, numerous studies around the world [10–13] have proposed project-based learning (PBL) as the most suitable means of achieving effective competence-based education [2,12,13] that integrates knowledge, skills and values. The models integrating PBL have their scientific basis in generating learning processes [10]. They are grounded in the belief that humans construct new knowledge over a base of what they already know [2] and of what they have experienced. This information is shared through active participation and interaction with others.

The Bologna Agreement, launched with the Bologna Declaration of 1999 (signed by 29 European countries), is one of the main voluntary processes at the European level (today, it has been implemented

in 48 states) that define the European Higher Education Area (EHEA). The Bologna Agreement had many positive impacts on sustainable planning education [4,9,14,15], opening up new opportunities that were closely linked to the international framework. In the context of the Bologna process, European universities have designed educational strategies from a competence-based approach to ensure that efforts to raise the quality of teaching and research go hand-in-hand with improving opportunities for under-represented groups [16]. One of these is the Erasmus Mundus program, which enables students to combine degrees from different disciplines to develop planning competences [8].

In many Master's of rural planning programs, sustainability and social dimensions include both a broad academic research and a professional field. Many universities have begun to introduce curriculum innovation and change to facilitate the curricular integration of generic skills underlying education for sustainable development. However, the literature and research in this area to date show few successful examples of comprehensive large-scale curriculum change and even fewer in relation to project planning of rural development.

As a professional field, sustainable planning is an institutionally embedded practice; it is also a practice that is inevitably interwoven with politics and with on-going conflicts over the allocation and use of public and private resources and public lands. Thus, politics is institutionally embedded as well. It follows that the activity of planning is understood and practiced differently in different institutional settings that vary significantly across countries and even cities. Moreover, within any given setting, planning must continuously reinvent itself as circumstances change [17].

There is universal acknowledgement that a wide-range of skills and knowledge is required to create action-orientated sustainability [18]. This international context requires the adoption of a "one-world" approach to planning education that equips students to work in different "world contexts" [8].

In the European Union, this professional field emerged with the new sustainable planning approach of the European LEADER Initiative, based on the concept of "endogenous development" [19] and recognizing the importance of local processes and social participation [20]. Numerous researchers have been described [21–24] the specifics of this new experimental sustainable planning approach, creating new rural development planning models. "Working with People" (WWP) is one of these models, and it is understood as the practice of the pre-professional team that seeks to connect learning-knowledge with action, from experimental learning by connecting to real-life situations, through common projects, which, in addition to the technical competences, incorporate capacity building and the value of the people who are involved and participate within the context of the project-based learning. In this WWP model, the concept of "social sustainability" confirms that strong social networks and social cohesion can be more important for a rural community's resilience than the project's physical structure [25].

Currently, in the sustainability planning debate, there is a consensus that environmental planning is different from sustainability planning, in that environmental planning is a component of sustainability planning. In addition, there are often strong links between social issues and environmental ones [26,27].

These papers explore various aspects of the social dimension and sustainability in the rural development context, and contribute to the global common good, through the world-class universities framework. The graduate level includes an Erasmus Mundus Master's (International Master's of Sustainable Development), in collaboration with six universities in the EU—Agropolis Montpellier CNEARC (France), Wageningen University and Research Centre (Holland), The Royal Veterinary and Agricultural University KVL (Denmark), University of Cork (Ireland), and University of Catania (Italy)—and another nine universities outside the EU, as well as a doctoral program adapted to the Bologna Agreement. These relationships favor the external mobility of students and academics, creating the Agris-Mundus Sustainable Development Alliance. The general objective of this program is based around validating the competence of individuals with respect to their knowledge, experience, and attitudes in relation to project planning for sustainable rural development [28].

This Master's Program at the Universidad Politécnica de Madrid (UPM) is officially recognized by the European Union and is fully integrated in the European Space for Higher Education. Characteristics

of the program are thus enriched with the criteria emanating from the Erasmus Mundus program: cooperation and mobility within higher education in order to achieve the objectives of improving European higher education and promoting intercultural understanding through cooperation with non-member countries.

Until this program was launched, Friedmann's theories of planning pedagogy and social learning [29–31] had never been applied to the European context of project planning for sustainable rural development. This was our main contribution at the time.

The general objective of this research is to analyze the way in which a PBL strategy combines project-program management for sustainable rural development competences across the implementation of the Bologna Agreement in the European Higher Education Area (EHEA).

It is based on the following hypothesis: Project-based learning (PBL) from the "Working with People" approach is a pre-professional practice that seeks to connect learning-knowledge with action—connecting experimental learning to real-life situations—and can contribute to a balance in the development of skills from the three dimensions (technical, contextual, and behavioral) associated with sustainability.

To analyze this hypothesis and determine the elements that the PBL strategy, we proceeded to study an International Postgraduate Program for Sustainable Rural Development—Agris Mundus Alliance for Sustainable Development—from the international competences of project-program management for sustainable development, according to the International Project Management Association (IPMA) standards.

In the following sections, we present the fundaments of a cooperative education methodology that is designed to prepare sustainable planners within the European Higher Education context.

## 2. Educational Strategy: Project Planning for Sustainable Rural Development

We used the prevailing PBL orientations in the teaching of sustainable development and the EHEA experience to advocate a methodological change—educators as role models and learners experiential learning by reconnecting to real-life situations; holistic thinking, skills, and knowledge associated with complex, multi-layered, and interconnected systems; interdependency and transdisciplinary connections between subjects; and approaches to developing and honing critical thinking.

As planning education varies so much over the world, reflecting each country's specific planning practices, any statement on the "core curriculum" of European planning education must pay respect to these variances and, therefore, cannot and should not be explained in too much detail. Nevertheless, we adopted the core requirements as guidelines—common competences and values shared by the international community that are suitable for high quality sustainable project-program management education all over the world [28]. The most suitable way to implement the change was to adopt a professional point of reference that would represent the needs of society. Thus, the International Project Management Association (IPMA) standard was adopted for learning and evaluation toward international competences of project-program management for sustainable development. The IPMA is an organization made up of more than 20 national professional associations from around the world. The adoption of this international standard enabled the initiation of the university stage of specialized training for future professionals, thus providing them with more opportunities to work in different "world contexts" [8].

### 2.1. How is Learning from Experience Possible?

The teaching activity depends on the type of teacher we are, and we say that there are two kinds of teachers—those who only transmit knowledge and those who create knowledge from their research. Professor Shulman from Stanford University said that, "Some teachers have 20 years of experience. Other teachers have one year of experience 20 times. If you think about it for a moment, learning from experience is a miracle" [32]. This challenge was the origin of our project-planning education strategy: "Learning from Experience" in the teaching of sustainable rural development.

The first steps of the PBL from "Working with People" were in 1987, with the emergence of an educational cooperation agreement between the Project and Rural Planning Department at the Universidad Politécnica de Madrid (UPM) and the Regional Government of the Community of Madrid [8]. This collaborative learning partnership was based on the concept of "planning as the professional practice" that specifically seeks to link knowledge to action. The educational framework emphasizes that planning is different from an activity like engineering where means are efficiently related to ends and projects determine the course of action [3]. The relationship between knowledge and action is interactive, a continual process of social learning among the actors involved. First, the sustainable planning team—the Department of Projects and Rural Planning of the University—is in an intermediate position between the sustainable rural project clients —the technical teams of the Head Office of Rural Development, Agriculture and Food of the Regional Government—and the beneficiaries of the projects—the population that lives in the rural areas. Second, the students are inserted into this framework to participate in a social learning process by solving real problems in teams [2]. During this process, students are enriched with external knowledge gained from direct contact with the different people involved (farmers, environmentalists, entrepreneurs, managers of local action groups, local development agents, and local-regional government managers).

This is not about articulating a theory and then applying it to a situation. The multi-stakeholder participation in the process of formulating rural development projects and adapting sustainable policies plans and programs for the Government of Madrid is fundamental for this work. It is a different way of thinking about planning the art of linking knowledge to action in a recursive process of social learning [30]. We emphasize that the knowledge we use and develop is not just the systematized knowledge of scientists, professors, and technical experts. The experiential knowledge in the course of the action [29–31] is equally important to the planning process.

The collaborative learning program—implementing PBL from "Working with People" approach—using the government–university agreement in the graduate and postgraduate curriculum introduced students to this new vision of planning and encouraged them to gain external knowledge from direct contact with diverse people. These professional relationships and complementary information enrich the students' base knowledge and lead them to develop new knowledge. This was one of the main elements of the strategy: participation in sustainable rural projects that respond to real needs, giving students the opportunity to leave the classroom to solve problems directly with external agents.

Until this program was launched, Friedmann's theories of planning pedagogy and social learning [29–31] had never been applied to the European context of project planning for sustainable rural development. This was our main contribution at the time.

The project-planning education process has a dynamic element in which students "learn to learn" about the reality of the rural world and how public administration—the project client—works. In the coordination of the project-based learning activities, a method of logic is applied in which learning experiences respond to structuring the methodologies of project planning and evaluation. The active method of "learning by doing" [32–34] is particularly relevant in projects and planning education and provides enormous potential for originality, creativity, and common sense. In addition, it is important to remember and to learn from the experience. It is an excellent method to develop the habit to remember even the most difficult experiences. Without that habit, little learning can occur [32]. Another feature of this approach is "uncertainty"—acknowledging that there are ranges of possible approaches to sustainability and that the situation is constantly changing, indicating a need for flexibility and lifelong learning [18]. Educational research studies increasingly demonstrated the need for student "ownership" of their programs as a basis for deep learning [32,35].

In this article, we draw directly on the research—Project-based Learning (PBL) from "Working with People" approach—that we conducted during more than 25 years at the GESPLAN Research Group at UPM, which complements planning and broadens the scope of postgraduate studies in the teaching of sustainable rural development.

## 2.2. Alliance for Sustainable Development: Agris-Mundus International Master's

In this way, teaching and research are included in an international strategy, which gradually provides students with competence training. Their knowledge increases, and their attitudes are shaped as they travel along this educational "road." They are given opportunities to acquire certain basic experience in advance. The graduate level includes an Erasmus Mundus Master's (International Master's of Sustainable Development), in collaboration with six universities in the EU—Agropolis Montpellier CNEARC (France), Wageningen University and Research Centre (Holland), The Royal Veterinary and Agricultural University KVL (Denmark), University of Cork (Ireland), and University of Catania (Italy)—and another nine universities outside the EU, as well as a doctoral program adapted to the Bologna Agreement. These relationships favor the external mobility of students and academics, creating the Agris-Mundus Sustainable Development Alliance.

The general objective of this program is based around validating the competence of individuals with respect to their knowledge, experience, and attitudes in relation to project planning for sustainable rural development [28].

This Master's Program at UPM is officially recognized by the European Union and is fully integrated in the European Space for Higher Education. Characteristics of the program are thus enriched with the criteria emanating from the Erasmus Mundus program: cooperation and mobility within higher education in order to achieve the objectives of improving European higher education and promoting intercultural understanding through cooperation with non-member countries.

The Master's Program has reinforced its international dimension (with an internationalization rate of 77%) through the international Alliance for Sustainable Rural Development. The international dimension of the program is reinforced in two ways: First, it forms a part of the international "Network of European Agricultural" (NATURA) related to sustainable rural development and created in 1988 in Louvain, Belgium. This association develops systematic actions within the field of development programs. Second, the program has reinforced its international dimension through Erasmus Mundus, establishing an association with eight higher education centers in non-member countries. Through this action, an increased global profile has been achieved, with a reinforced worldwide presence, and associations with higher education institutions in non-member countries have been created.

The research dimension is important in the Master's and offers direct access to the UPM's PhD program, complementing the experience with the participation of noted professionals from other universities, including Stanford (USA) and Berkeley (USA). On the other hand, in 2009, after the accreditation process, the program was incorporated within the Registration of Competence Development Programs, being the first registered program for the IPMA in Spain and the first in the world that applies IPMA competences to sustainable rural development [28]. According to the IPMA model, the individual balances the project's economic, social and environmental aspects to meet the requirements for sustainable development and to make the project results sustainable [36].

We work together with different institutions that support the master's program with conferences, field trips, and permanent agreements for research stays during the Master's thesis, etc. Many of our students conduct a research stay as part of their Master's dissertation. All tutors are recruited through the institutions participating in the Alliance for Sustainable Rural Development Master of Science Network (Table 1) partners. This Alliance uses research projects in rural communities in public and private lands as "laboratories of learning" for the Master's students.

**Table 1.** Alliance for Sustainable Rural Development Network partners.

| Academic Institutions | |
| --- | --- |
| Montpellier SupAgro (Francia) | UCAV–Universidad Católica de Ávila |
| Wageningen University and Research (Holland) | Pontificia Universidad Javeriana (Colombia) |
| The Royal Veterinary and Agricultural University (Denmark) | ECOSUR, El Colegio de la Frontera Sur (México) |
| University of Cork (Ireland) | Universidad Nacional de la Plata (Argentina) |
| University of Catania (Italy) | Universidad Nacional de Colombia |
| Colegio de Postgraduados de (México) | Universidad Nacional de San Agustín (Perú) |
| Universidad Politécnica Salesiana (Eucador) | Universidad Católica Sedes Sapientiae (Perú) |
| Universidad Nacional Mayor de San Marcos (Perú) | Universidad Autónoma de Nariño (Colombia) |
| Universidad Uniagraria de Colombia | Pontificia Universidad Católica Madre y Maestra (Rep. Dominicana) |
| **Professional Partners** | |
| International Project Management Association (IPMA) | GALSINMA, Grupo de Acción Local de la Sierra Norte de Madrid (Spain) |
| Asociación Española de Dirección e Ingeniería de Proyectos (AEIPRO) | National Park Service, Sierra de Guadarrama |
| Instituto Interamericano de Cooperación para la Agricultura (IICA) | International Center for Agricultural Research in the Dry Areas ICARDA Jordania |
| Food and Agriculture Organization (FAO) | IALA Guaraní–Instituto Agroecológico Latinoamericano (Paraguay) |
| Coordinadora de Mujeres Aymaras, CMA (Perú) | ANC–Associação de Agricultura Natural de Campinas e Região (Brasil) |
| FESBAL–Federación Española de Bancos de Alimentos | MAGRAMA–Ministerio de Agricultura, Alimentación y Medio Ambiente National Park Service |
| Fundación Ingenio (Spain) | AMSA–Autoridad para el Manejo Sustentable de la Cuenca y del Lago de Amatitlan (Guatemala) |
| IDC Cuenca–Instituto de Desarrollo Comunitario de Cuenca (Spain) | Coopérative Fermes de Figeac (Francia) |
| Zerca y Lejos ONGD (Spain) | |
| Asociación Agranda la Olla (Spain) | |
| Gobierno Provincial de Manabí (Ecuador) | |
| **Agency and Organizational Partners** | |
| CONICET, Consejo Nacional de Investigaciones Científicas y Técnicas (Argentina) Servicio Holandes de Cooperacion al Desarrollo | AECID, Agencia Española de Cooperación Internacional para el Desarrollo |

With the objective of disseminating knowledge, the UPM Research Groups carry out seminars and workshops with different approaches, including territorial research workshops in selected disadvantaged communities in Latin America and encounters for dialogue, reflection, and debate with entrepreneurs, academics, and civil society actors linked to different "laboratories of learning." Land resource managers and scientists regularly collaborate on issues related to rural development, social and environmental conflicts, sustainability, and preservation of cultural resources.

Two research groups from the UPM jointly developed this international academic program. First, the GESPLAN Research Group pools together a group of professors and researchers who have worked for more than 25 years in the area of project management and planning for sustainable rural development of communities. Among the members of the group are young researchers and professionals in the field of agronomic engineering, industrial engineering, and economics. This group of experts has extensive experience in planning in development, program evaluation, and project management and hence, the problems associated with rural development. On the other hand, the SILVANET Research Group is made up of professors, researchers, and staff from the School of Forestry and Natural Environment Engineering (ETSIMFMN, UPM), and its focus is ecology and sustainable environmental management, in particular the modeling and simulation of natural processes, forestry, landscape ecology, territorial planning, and management of the environment.

## 3. Methods

The study was conducted within UPM, in Spain, in a highly international context. Students (204 students) were sampled from 43 different countries using self-assessment questionnaires. Given

the holistic and diverse nature of program, we collect multiple types of data from different perspectives of competencies.

*3.1. Data Collection*

Mixed methods were used to collect and analyze data and integrate findings [37]. With a focus on staff perceptions, we sought to examine the development of competencies after program implementation. Each year, over the course of nine years, the methodology for evaluation included the self-assessment tools and evaluation workshop with the students and the all the staff. These instruments have been applied since 2009 to a total of 204 students from 43 different countries enrolled in the UPM Master's.

(a) The self-assessment questionnaire was designed based on the instrument used by the IPMA [36] for the evaluation and certification of project management competences, which includes 92 items—46 on knowledge of competence and 46 on experience—covering the following three areas of competence: 20 technical elements, 11 contextual, and 15 of professional behavioral, according to the IPMA's Individual Competence Baseline [36]. This instrument used a Likert scale [38] commonly used in social sciences to assess perceptions and qualitative aspects [39]. With a focus on students and staff perceptions [37,40], we sought to examine the implementation of the program in terms of whether the program was delivered as planned, competences developed in students, and the context that may have influenced the program. The self-assessment tools were created using Moodle (UPM's virtual platform). In order to unify the criteria according to the IPMA competence assessment, and in order to indicate the level of opportunities for improving skills, we use the same scale for each skill element covered by the strategic learning planning.

(b) The empowerment workshops are face-to-face sessions, lasting approximately two hours, and are organized through participatory techniques that allow participants to quickly and dynamically discuss the results of the evaluation of each subject and the Master's degree as a whole [40]. This final empowerment workshop takes place with the objective of completing the process of continuous evaluation and to collectively discuss and contrast the evaluations carried out individually.

*3.2. Data Analysis*

The IPMA ICB dimensions used to analyze and present the results were organized in three competence areas:

1. People: define the personal and interpersonal competences required to succeed in sustainable projects and programs;
2. Practice: define the technical aspects of managing sustainable projects/programs;
3. Perspective: define the contextual competences that must be navigated within and across the broader environment [36].

In the next sections, we present our findings and offer the Master's as a model for educating students on concepts related to the planning of sustainable rural development and sustainable management.

## 4. Results and Discussion

*4.1. Overall Results of the Master's Program*

From 2009 to 2019, 225 students from 43 different countries and from very different educational backgrounds have taken part in the program. The novel and adequate design of the program has allowed a history of numerous international students, being the most international program at UPM. The highlights of the high internationalization rates of enrolled students in this Master's Program are shown in Table 2.

**Table 2.** Rate of internationalization in the Master's program.

| | 2009–2010 | 2010–2011 | 2011–2012 | 2012–2013 | 2013–2014 | 2014–2015 | 2015–2016 | 2016–2017 | 2017–2018 | 2018–2019 | 2019–2020 |
|---|---|---|---|---|---|---|---|---|---|---|---|
| Pre-registered students (Number) | 167 | 108 | 116 | 135 | 82 | 88 | 108 | 82 | 97 | 50 | 93 |
| Pre-registered international students (%) | 44.9 | 32.9 | 58.6 | 56.3 | 68.3 | 63.6 | 69.4 | 83.2 | 92.8 | 84 | 74 |
| Enrolled students (number) | 20 | 16 | 18 | 19 | 21 | 26 | 25 | 22 | 15 | 22 | 21 |
| International students enrolled (%) | 25.0 | 31.2 | 55.5 | 73.6 | 85.7 | 88.5 | 86.7 | 90.9 | 86.7 | 81.8 | 61.9 |

The joint work between the stakeholders concluded in the design of the study program (Table 3) in 16 modules during the winter and summer terms: nine are compulsory core modules with a total of 35 European Credit Transfer and Accumulation System (ECTS) credits; six are electives (13 ECTS), and students need to attend at least 10 ECTS out of the electives; and the annual dissertation of the Masters' Project (15 ECTS) is compulsory and takes place in both terms. The ECTS credits are a standard means for comparing the volume of learning based on the defined learning outcomes and their associated workload for higher education across the European Union and other collaborating European countries. This grading scale has been developed to provide a common measure and facilitate the internationalization and mobility of students among European institutions of higher education.

**Table 3.** Degree program and the students' workload of the Master's Postgraduate Degree in Project Planning of Rural Development and Sustainable Management, https://desarrollorural.us.

| Learning Sections | Core Modules | Elective Modules |
|---|---|---|
| **I** Planning, Management and Evaluation of Sustainable Rural Development (21 ECTS) | Planning in the Public Sphere: From Theory to Action (5 ECTS) | Rural Planning Based on Ecology: Introduction to Development Models (5 ECTS) |
| | Evaluation Design: Methods and Tools (5 ECTS) | Participatory and Collaborative Evaluation for Empowerment (2 ECTS) |
| | Project Management for Sustainable Rural Development (5 ECTS) | Competence baseline for the Competences in Project Management (2 ECTS) |
| **II** Quantitative Techniques for Sustainable Development (9 ECTS) | Methodologies in for Territorial Studies: Remote Sensing and GIS (3 ECTS) | Applications for Socially Intelligent and Complex Systems in Rural Development (2 ECTS) |
| | Support Systems for Decision-Making (4 ECTS) | |
| **III** Sustainable Management of the Environment and its Biodiversity (12 ECTS) | Techniques and Models for Sustainable Management and Conservation of Biodiversity (4 ECTS) | Network of Natura 2000. Concepts and Evaluation of its Conservational state and its Habitats (2ECTS) |
| | Socio-Economic Assessment of Sustainability (2 ECTS) | |
| | Sustainable Management of Soil and Water (4 ECTS) | |
| **IV** Research in Sustainable Rural Development (6 ECTS) | Scientific Writing: Editing the Thesis and scientific articles (3 ECTS) | Human Development (3 ECTS) |
| Module **V** (15 ECTS) | Annual Dissertation of the Master's Project (15ECTS) | |

The analysis and reflection regarding proposals and conclusions from this process enable the creation of a series of "lessons learned." The results combines project-program management for sustainable rural development competences organized in three dimensions of competences (technical,

personal, and contextual) according to the IPMA model and covers the social-relations system (politics, public-administration, private-entrepreneurship, and civil society) as a synthesis of societal models [3,41]. Table 4 summarizes the results of Self-assessment skills in the 2009–2019 period (at the beginning and end of the academic year) of knowledge and experience acquired by students after project-based learning (PBL) from the "Working with People" approach, based on competencies for sustainable development project management [36].

**Table 4.** Overall results of the self-assessment competences (knowledge and experience).

| Dimensions | Baseline | Final | Variation |
|---|---|---|---|
| Improvement in technical skills | 38% | 83% | 45% |
| Improvement in behavioral skills | 58% | 92% | 34% |
| Improvement in contextual skills | 23% | 75% | 52% |

*4.2. Implications for the Different Stakeholders and Emerging Good Practice from the Three IPMA Dimensions*

The findings identified and emerging good practice from the PBL strategy, and according to the IPMA model, can be categorized as follows: (1) Personal and interpersonal competences; (2) Perspective—holistic thinking and intellectual coherence and (3) Practice—connecting to reality.

4.2.1. Personal and Interpersonal Competences: Developing "Global Skills"

As mentioned in all empowerment workshops, one of the greatest values of the Master's is to offer an international ecosystem, which is especially suitable for multicultural learning and developing "global skills." The IPMA competences (technical, contextual, and behavioral skills) have become the central component on which the PBL strategy is built and represent a tool that can be used by sustainable project management professionals to become globally competent and improve international employability [28]. Faced with this challenge, the PBL strategy define "global competency" as the knowledge and skills people need to understand today's world and work in the fields of project-program management for sustainable rural development. According to the IPMA, individual competence is the application of knowledge, skills, and abilities in order to achieve the desired results [36].

The results of self-assessment skills in the 2009–2019 period (Table 3) show that, using the PBL methodology, students improve knowledge and professional skills in three sustainable project management areas (technical, contextual, and behavioral). This highlights the improvement in the contextual skills dimension (52%). This contextual dimension is the most closely related to sustainability and has the objective of designing and understanding the strategic processes so that a project (or program or portfolio) is formulated and managed by considering the contextual aspects and generating long-term benefits that guarantee sustainability. This dimension addresses the need for a more holistic and coherent vision of projects in terms of the degree of adaptation to their environment and society in general [16,42]. In this contextual dimension, the projects are analyzed, ensuring that they are highly correlated with the mission and the sustainability of the organization. They also ensure that individuals understand the Health, Safety, Security, and Environmental (HSSE) regulations that are relevant to the project; it identifies and ensures that the project complies with the relevant sustainability principles and assesses the project's impact on the environment and society [36].

Within the technical skills area (with an estimated improvement of 45%), the temporal dimension for the effective integration of sustainability into the different phases of the project cycle is taken into account [42,43]. As in other experiences [27], these concepts are included to analyze not only the life cycle of the project processes but also the life cycle of the resources used and the effects (products) caused by a project.

Behavioral skills (human dimension) are a key issue in PBL strategy (with an estimated improvement of 34%). Among the important behavioral skills for sustainable development planning, teamwork is considered a transversal competence, a central topic of the knowledge and skills

required [37]. From the annual empowerment workshops, students highlight the opportunity to become globally competent from a highly international and cultural teamwork context. Sustainable development planning and project management are established as the pre-eminent methods for implementing changes, but it is the people who are leading the way for changes in the world. Reimers recognizes behavioral global competencies are also the attitudinal and ethical dispositions that make it possible to interact peacefully, respectfully, and productively with fellow human beings from diverse geographies [44].

The IPMA standard reinforces the international PBL strategy with an emphasis on understanding key elements of sustainability [45,46]. These global competencies and the certification of people using international standards have become important elements in the PBL strategy, verifying the competence of an increasingly mobile and global workforce. In response to this growing need, ISO (International Organization for Standardization) aims to harmonize the various procedures used around the world for certifying the competence of personnel in occupations or professions [47]. The adoption of this standard enabled the initiation of specialized training open to the world, making it the most international Master's at UPM (Table 2)—80% international students—with effects on one of the great challenges of teaching: the cultural and contextual implications in crossing borders [48] and in consolidated knowledge within the framework of the epistemology of planning, projects, and sustainable rural development [49,50]. As Kunzmann and Yuan note, "teaching foreign students requires experience, sensibility, and an understanding of cultural differences. It also requires time and patience" [1] (p. 69).

As in other experiences [51,52], all the activities reinforce the globalization of knowledge in the sense that different disciplines are incorporated—economics, sociology, agrarian policy, environmental sciences, territorial planning, engineering projects—consistent with the needs that arise as contextual competences of project/program management. As a result, teaching modules are not treated in isolation but rather present a relationship among the three dimensions of competence—technical, behavioral, and contextual [8]. With this PBL approach based on international global competencies, the Master's program prepares students for professional life in the real world in order to be able to face global challenges from projects. According to employability data from the last four editions of the master, 96% of the graduates of the Master's are currently working, both in public and private entities, on topics related to the program (rural development, global poverty, solidarity, international consulting, research and development, engineering projects, training, human rights, and environmental and territorial planning). Of course, in relation to this behavioral skills dimension, structural problems were encountered, mainly for coordinate cooperative learning activities and group interdependence.

### 4.2.2. Perspective-Contextual Dimension: Holistic Thinking and Intellectual Coherence

In the Master's Program, one of the aspects most valued, both in national and international accreditations, is this holistic approach, with a novel intellectual coherence, encompassing an open-ended exploration of interdependency and transdisciplinary connections between subjects around three topics. There is universal acknowledgement that a wide range of skills and knowledge are required for education relating to sustainable development projects [9,18]. The sustainability issues are related to planning and policy development [52], certification, policy assessment, project sustainability, ethics [18], equity and empowerment [41], and what it implies about the way our holistic thinking is conducted. In the Master's Program, the teaching modules (Table 4) include technological aspects, but also environmental, social, and economic considerations, as well as policies relating to planning sustainable development projects.

In the Master's Program, the skills and knowledge for sustainable development are mainly associated with the intellectual contributions of three professors and also include common research and teaching approaches to develop and hone critical thinking. These professors belong to the group of relevant innovative people within the planning and projects domain who are able to create new ways of thinking and acting within their university departments [8].

Experience shows that, even within Europe, schools of planning do not agree on which competences professional planners should have. Planning is still understood quite differently from country to country and from dominating discipline to discipline, even though planning education aims to teach students how to plan "for people" [1].

In our case, adapting Friedmann's planning theories to practical pedagogy is not only understood as working for others but as working "with others" [19]. From this social dimension and perspective, the Master's has the mission of providing participants with a specialization in subjects and content that is directly related to professional activity across three bodies of knowledge: planning, projects and sustainable management.

With regards to planning as a scientific meta-discipline, John Friedmann is one of the most influential worldwide figures from the second half of the 20th century. His contributions to planning theory and practice have yielded a rich harvest both in the context of sustainable development and planning education [53–55]. The PBL strategy model in the context of the Master's has been applied to several real experiences adapting Friedmann's ideas and empowerment [41] theory to practical pedagogy in sustainable rural development [18].

On the other hand, the PBL strategy adopts a conceptual framework from sustainable planning in Spain under the intellectual drive of Angel Ramos and Ignacio Trueba. They initiated the Project and Rural Planning Department at UPM in 1985 and were pioneers in the field of project planning and sustainable management in engineering. From the very beginning, they contributed to the creation of the Spanish Association of Engineering Project Management (AEIPRO) and promoted the professional standards and core requirements for project planning management.

Both teachers saw project planning as a transformational tool that required a different approach from what prevailed during the 1970s in planning across both public and private domains. They also called for structuring knowledge and action in a different way, both in academic institutions where it helped to bring about change and with direct action through projects [8]. Professor Ramos' scientific research contributions and his colossal sensitivity and respect for nature focused on sustainable management and environmental awareness in the management of projects and the value of the environment as a resource for planners and public managers. Regarding social integration in the sustainable project, Professor Trueba's experience and knowledge, including his contributions to the design of new methodologies for projects planning, were integrated into the knowledge of the Master's [56]. This allowed us to establish a close link between engineering teaching—until then merely technical—and the sustainable management of international organizations' large initiatives, as well as the link between the mission of technique and engineering and their role in finding solutions for the great problems facing humanity, such as the fight against hunger and poverty in rural areas. Highlighting this social integration, Baldwin and King [25] include the concept of social sustainability in their research titled "What About the People?," confirming that strong social networks and social cohesion can be more important for a community's resilience than the physical structures of a project. This social orientation of sustainability is a hallmark of our master's program and examines the challenges of scaling up participation in sustainable development projects, strengthening the skills of weaker stakeholders, and creating organizational cultures that support participation [57–61].

From the previous holistic thinking and intellectual coherence, research-training and knowledge transfer are related in the field of sustainable rural development (Table 4) to prepare sustainable planners in the Bologna European education context. The PBL strategy framework, from the WWP approach, integrates relationships between research groups, educational innovation, and collaborative partners for the construction of sustainability from three interrelated dimensions (research activity, teaching-learning, and collaborative partners for sustainability), as shown in Figure 1.

Teachers are members of three recognized and regulated work structures at UPM—the GESPLAN Research Group, the Research Group for Sustainable Management (SILVANET), and the Educational Innovation Group (GIE-Project)—and many teachers and researchers belong to both structures, facilitating teaching–research integration within a learning strategy based on sustainable project

management competences. This relationship between Research Groups and Educational Innovation Groups represents an interdisciplinary and international team involved in the Agris Mundus Sustainable Development Alliance, from different departments with different specialties (agricultural engineering, forest engineering, environmental sciences, civil engineering, industrial engineering, technical architecture, public works school).

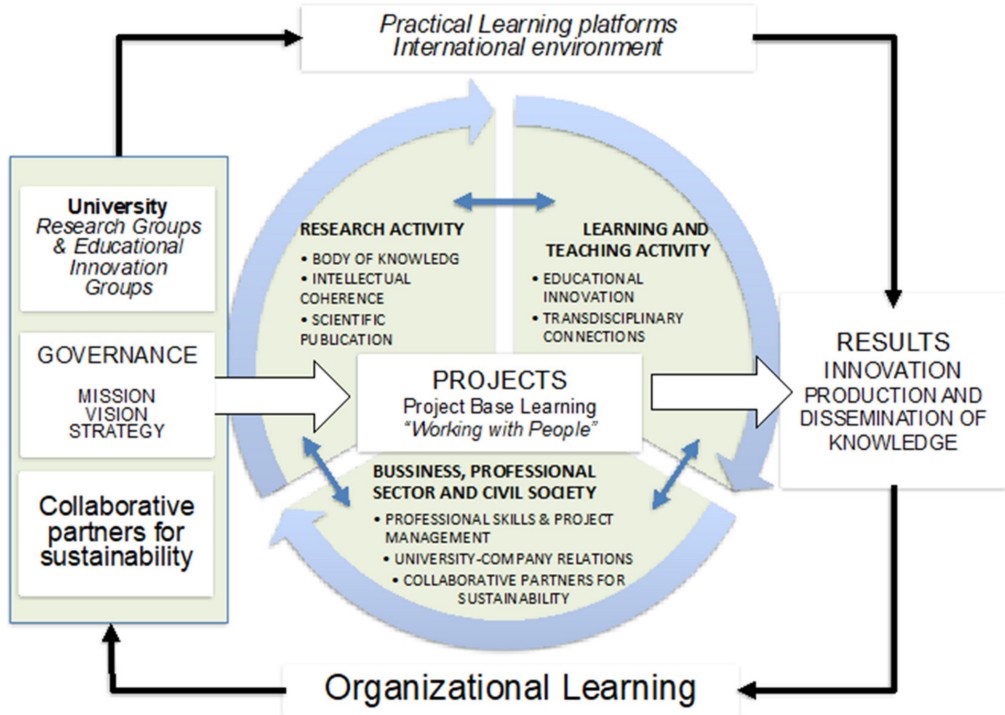

**Figure 1.** Perspective-contextual dimension: project-based learning (PBL) strategy.

The opportunities for interdisciplinary and international cooperation allow the development of individual skills, through PBL—thesis project and Master's thesis—as an appropriate educational tool to generate professional experience that strengthens experimental learning [33] and relations with the collaborative partners and local community, dealing with real-life problems and experiences, developing the ability among students to enact social changes, links with governments, environmental management institutions and higher education.

This PBL strategy is also based on the implementation of one Educational Innovation Program (EI-Program) for promoting global and international project management skills. The EIP originates from the policies adopted by the UPM to promote "Educational Innovation Groups (EIG)". The EIG-Project was officially approved in 2005 as a group set up by people who demonstrate research experience, training and projects of sufficient consistency in the fields of engineering and professional sustainable project management skills [28,62].

At the core of the PBL is the conceptual framework "Working with people", which synthesizes the intellectual connection and achieves a new approach for the planning of sustainable rural development projects [19,63,64]. This "Working With People" conceptual model is understood as the professional team-practice that seeks to connect knowledge and action with a common project, which besides the technical value of production—of goods and services—mainly incorporates the value of people who get involved, participate and are developed through the actions carried out within the context of the sustainable project [19]. The WWP model is the result of 30 years of experience in teaching and research in the subject of sustainable rural development project planning from the GESPLAN, "working with people" in several European contexts and emerging countries.

The social system that surrounds sustainable development projects is to cover people's conduct and moral behavior and it sets out the "foundations" to make people (both from private and public fields) come to work together, with commitment, confidence and personal freedom.

Project stakeholders is one of the most important knowledge areas in this PBL strategy, as project success is measured based on stakeholders' satisfaction, which can vary according to different perspectives [65]. The production and dissemination of knowledge, from the projects results, are one of the central foundations of the PBL strategy, and the commitment of the university.

In this process, behavioral competencies, with ethics and values, are fundamental and appropriate elements to overcome potential moral conflicts related to the parties involved in the project [36]. These personal and interpersonal competences are certainly important for sustainability integration in projects [27,66] for understanding and providing guidance on ethics and moral values and are related to corporate policies and practices. The model therefore seeks to enhance governance through project management, from ethical and moral standards, from the university [67,68].

### 4.3. Practice: Experiential Learning by Reconnecting to Real-Life Situations

From this "practice dimension" the results focus on real and practical life issues and actual experiences as learning situations. For this purpose, the implementation of the model—as a guideline in the field of sustainable development planning—has led to different applied research, which has been carried out with projects developed from the so-called "Chair Company". Through these alliances, the university can play an active role in sustainable development, promoting innovation in society [69]. In the UPM, this Chair-Company relationship is a means to establish a long-term strategic partnership between the University and an external institution (civil society, public body, or private company) in order to carry out training-research and knowledge transfer in areas of common interest. Through this chair company, the PBL puts students in touch with real-life problems and real experiences to find real "solutions" in the territories for people.

As Siemiatycki states, over the past 50 years, a recurring theme in planning scholarships has been to comprehend and categorize the roles, epistemologies, and dilemmas commonly faced by the planning practitioner in society [70]. With these goals in mind, our learning and teaching strategies are focused on project-based learning (PBL) as the most suitable means of achieving effective competence-based education, fostering holistic thinking/acting, and encouraging teamwork and leadership [2]. Figure 2 summarizes these chair company partnerships, which are the means to generate collaborative projects and "practical learning platforms" used in the Master's.

Thus, it is about a new educational approach based on "learning" under the interaction of the students with real-life case studies and real projects. This environment—chair company partnerships—provides the opportunity for both action and reflection, showing that learning is both an active and reflective process, and also provides the opportunity to include research in the field of sustainable rural development project planning and related fields.

In both accreditations and the empowerment workshops, these chair company partnerships are highly valued in the PBL strategy aims to teach students how to plan "with others" and "with people," combining academic rigor and researchers with practical relevance and thinking, acting together to solve the problems facing society. We have called our learning, teaching and research model "Working with People" (WWP), and it is understood as the practice of the pre-professional team that seeks to connect learning-knowledge with action, from experimental learning by connecting to real-life situations, through common projects [8].

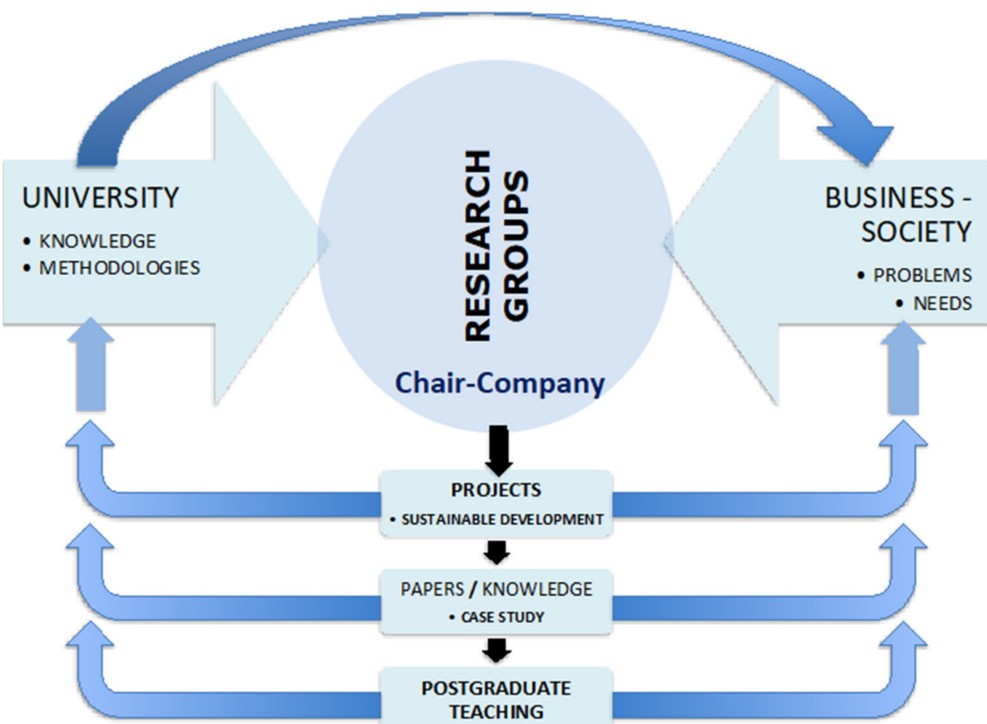

**Figure 2.** PBL strategy.

Sustainable planning requires continuous communication while "working with people" such as politicians, stakeholders, and target groups. This PBL from "Working with People" approach puts an emphasis on small "action groups" and inter-personal dialogue among multi-stakeholders, capable of providing a moral foundation for planning, similar to what Friedmann suggested [53]. Planning students wishing to work in sustainable planning practice in their country have to learn how to communicate with people and stakeholders, with politicians and real estate managers. The PBL–WWP is integrated into international discussions on social learning and it incorporates key elements from planning as social learning models [55], collaborative participation theory of planning, international project management models [36], and challenges of educational research, especially those that integrate behavioral and contextual competencies from the perspective of professionals and stakeholders [71].

Below are some of the "practical learning platforms" used in the Master's. These are implemented by the "Chair Companies" managed by GESPLAN and use rural communities and lands as learning laboratories for students to build university-company-society relationships, encouraging the implementation of sustainable projects and programs.

A.  Social innovation and sustainable rural development: One of the "practical learning platforms" used in the Master's focuses on establishing agreements with local action groups from the LEADER European Initiative within the EU's sustainable rural development framework. These agreements have provided the students with the opportunity to work on real projects with local organizational structures, allowing them to understand real problems and experience planning as social learning [20–23] in the context of innovation and rural development in Latin America and Europe [2,24]. In these platforms the "planning as social learning" body of knowledge [53,72,73] is especially considered. Innovation as a process of social learning is understood as an open and interactive process with an important social dimension, which means a constant adaptation of the forms of knowledge and learning to the constantly changing market and technological conditions [64].

B.  Sustainability of food production systems and rational consumption: Another "practical learning platform" is the UPM Chair-Food Banks (FBC) as a means to establish a long-term strategic

partnership in order to carry out training-research and knowledge transfer on issues of food sustainability and rational consumption. This agreement and the idea arose at the Spanish Federation of Food Banks (FESBAL), who reached out to the GESPLAN. The activity took place in Spain, with 55 Food Banks from a total of 254 food banks in Europe. There are three main areas of learning in action for students: The Rational Sustainable Food Consumption (CORAL) program with schools, campaigns to raise awareness with social networks, and practices in companies on issues related to sustainability of food production systems and rational consumption [74,75]. After six years of activities, the results have been very positive and have reached more than 2500 pupils in 25 schools in different Spanish cities. Another axis of the actions taken consists of a series of activities around the university, primarily including awareness campaigns against food waste, aimed at university students to spread the message of rational consumption.

C.  Entrepreneurs for the sustainability of rural areas (JESTER): Another "practical learning platform" is established by the Fundación Tatiana Pérez de Guzmán el Bueno (FTPGB) Chair Company, a private Spanish foundation. This Chair-Company consists of the planning and implementation of a sustainable entrepreneurship program called "Young entrepreneurs for the sustainability of rural areas" (JESTER Program). This program's objective is to identify potential entrepreneurship resources in order to promote entrepreneurship projects with a sustainable and environmentally friendly approach [76]. Each year, the Master's students participate in the program, holding work meetings with entrepreneurs. In addition, Master's thesis projects related to the sustainability of the enterprises are developed. The elements that characterize the program are the result of participatory collaboration between the different stakeholders involved. Within this collaboration framework, it has been possible to generate an entrepreneurship strategy in which the participation of local people and the teaching-research relationships have played a key role [77–79].

D.  Sustainable livelihoods for small-scale producers and sustainable approach of land governance: Another "practical learning platform" is developed from a strategic collaboration between the FAO and GESPLAN. This agreement has developed knowledge and skills regarding the Principles for Responsible Investment in Agriculture (RAI principles) and the Voluntary Guidelines on Responsible Governance for Land Tenure (VGGT). These RAI principles aim to contribute to sustainable and inclusive economic development and the eradication of poverty (Principle 2) and to conserve and sustainably manage natural resources and increase resilience (Principle 6). The VGGT refers to the need for a "holistic and sustainable approach" in land governance [80]. The work with universities, coordinated from GESPLAN, aimed to insert RAI principles and VGGT guidelines in university curricula and research and reached 27 universities from seven countries (Peru, Argentina, Ecuador, Mexico, Colombia, Dominican Republic, and Spain) [81,82]. With the participation of Master's students, five research case studies (carried out in Mexico, Peru, Colombia, and Chile) were delivered to the FAO, acknowledging the great potential for the RAI principles [83–85] and VGGT [86,87] to contribute to sustainable and inclusive economic development and for a sustainable approach in land governance.

E.  Project-based governance framework for sustainability: Finally, other knowledge and skills have been developed from the Fundación Ingenio Chair Company. In this "practical learning platform," planning and sustainable management models with bottom-up approaches are explored, incorporating a holistic approach stemming from pursuing economic, social, and environmental sustainability [19]. In this context, a rural development project has been implemented, and it is managed by the Aymara women's organization (CMA) in Puno, Peru. This project involves a situation in which groups of female associates are developing skills to turn their craft activity into a successful sustainable business project [88,89]. In addition, in this Project-based Governance Framework for sustainability, the activity of Camposeven agri-food cooperative is developed. The Camposeven agri-food cooperative is located in the region of Murcia (Spain) and produces vegetable organic and biodynamic products crops [90,91], grown

both in open-air and in greenhouses. Throughout this "practical platform," a conceptual proposal has been developed to address sustainability in agri-food cooperatives, from project-based governance [92] and in the construction of alternative pathways in sustainable agriculture [93,94]. The Master's students participate in the program each year, holding work-meetings with farmers from Camposeven. In addition, there is an annual meeting in the Master's with the president of the Aymara women's organization (CMA), who travels to Spain every year. Over the years, numerous students have completed their Master's thesis within the framework of both platforms [94–96].

## 5. Conclusions

The university, and especially world-class universities [16], can play an active role in promoting sustainability and innovation in society [97]. From the implementation analysis, it is clear that project-based learning (PBL) from the "Working with People" approach, based on practical learning platforms, is an appropriate strategy to explore sustainability issues and solve society's problems.

As in other research [98–102], the growing importance of social learning in the science of sustainability is highlighted. This experience explores various aspects of the social dimension and sustainability in the rural development context, through planning as social learning and the Working with People (WWP) framework [19], combined with theories of social learning and empowerment, contributing to sustainability. While the document focuses on planning of rural development and sustainable management projects, the model can be applied to other postgraduate programs.

The transactive planning [29] and project-based learning approach both involved processes of mutual learning and are closely integrated with a capacity and willingness to act. These concepts are grounded in the belief that people construct new knowledge over a foundation of what they already know and have experienced. That is why Friedmann's theories, with contributions in planning as social learning, are particularly suitable for preparing "sustainable planners" in higher education institutions and in the world-class university context [103–108].

This experience provides evidence that the process of social learning within sustainable development can be effective for different projects implemented by the public and the private sectors, connecting teaching and research with the needs and problems of the real world.

The results show that project-based learning (PBL) from "Working with People" approach has been highly effective, highlighting the following elements:

A. The importance of coordination and the role of research groups in the design of the strategy for intellectual coherence and strategic vision on the pillars that shape world-class universities [3,4]. One of the great values of these research groups is to offer an international ecosystem that is especially suitable for multicultural learning to allow students to become globally competent professionals. The "practical learning platforms"—created by the Chair-Companies and managed by the research groups—demonstrate the correct approach of the PBL strategy based on teaching-research linked to the professional sphere.

B. The methodology fosters a research spirit and innovation for the generation of new knowledge, productive thought, and motivation to learn and solve problems [109]. Debates take place with postgraduate research students from different countries and cultures in order to improve understanding between different perspectives of sustainability, planning traditions and cultures.

C. The necessary competences approach advocated by the EHEA have been used as an opportunity to establish a new connection with the professional world and adopt the professional standard recognized internationally as our point of reference. This connection also permits linking university education with a system of professional certification, which opens up better future opportunities for UPM's planning graduates. The future of sustainability planning is dependent upon the education of an interdisciplinary workforce with broad and holistic skills related to environmental, social and economic stewardship [9]. The balance between the three dimensions

of competences—technical, contextual and behavioral competence—and the integration of social aspects are crucial for sustainability.

Throughout the course of the activity, this strategy has been shown and discussed in numerous international symposiums showing the University as an "engine for the transformation of society" [110]. It has a dual and ambitious objective: to show the path that some worldwide universities [16,111] have traveled through postgraduate experiences in relation to sustainability; and to encourage improved governance over the noble task of serving society more effectively by linking research, teaching innovation and personal development, focused on promoting sustainable development and social progress within the university.

**Author Contributions:** The individual contribution and responsibilities of the authors are listed as follows: A.C.-M. and I.d.l.R.-C. designed the research, conducted model validation, and wrote the first draft of the article. J.I.P. revised the manuscript, provided some comments, reviewed the relevant literature, and helped edit the manuscript. All authors reviewed and approved the final manuscript. Professors A.C.-M., expert on sustainable development planning, and I.d.l.R.-C., expert on sustainable project management field, headed the research team. Professor J.I.P., expert on rural development and sustainable management in training and research programs, is doing his PhD under the supervision of professor A.C.-M., and this is a main contribution of his research, focused on the model to insert RAI principles and VGGT guidelines in university curricula in both teaching and research.

**Funding:** This research was funded by European Commission (grant number 05-A1-2006) and by UPM Chair Companies: Fundación Ingenio, F. Tatiana Pérez de Guzmán el Bueno and Chair Food Banks FESBAL.

**Acknowledgments:** The authors acknowledge the Agris Mundus Alliance for Sustainable Development (Erasmus Mundus Master—International Master's of Sustainable Development). We also thankfully acknowledge support from Master´s students, professors, researchers, and staff from the GESPLAN Research Groups and from the SILVANET Research Group. The present work has been done under the umbrella of a PhD research thesis on the sustainable development planning field.

**Conflicts of Interest:** The authors declare no conflict of interest.

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
