# Peer review of "Sustainable Development Planning: Master’s Based on a Project-Based Learning Approach"

_sustainability, doi:10.3390/su11226384_

Round 1

Reviewer 1 Report

Dear authors,

The topic of the article is interesting and the theoretical framework is well defined and described. References are wide and inclusive.

Nevertheless, strong improvements about research methodology, results and implications are required.

1) From the beginning of the article, authors need to specify that this is a case study which includes details and finding only based on one single case and context: one program, from one university. It is impossible to generalize this experience to other environment, since authors did not replicate the study to other masters, universities, countries.

2) It is a descriptive research, no theories will come from the findings. 

3) Research methodology needs to be better described and results need to be presented in a clearer and more objective way through, for example, one or more tables.

4) Authors need to specify which are the research objectives defining one or more research questions or research hypothesis.

5) These questions or hypothesis need to be recap in results presentation, implications and conclusions.

6) To many details about the actual case are explained. All these details are just a description of the Master Project and are not useful for the paper since they make it impossible to repeat the research to other situations. The not repeatability of the study creates a lack of relevance to the research

7) All the results' chapter is too long and mixes up results, theoretical framework, information about the project, etc. I advice to create a short chapter about the actual numerical and objective results, one chapter about implications of the results for the different stakeholders (students, teachers, ...) and finally one chapter with final conclusions, research limitations and future research.

Best regards,

the reviewer

Author Response

Dear reviewer,

Thank you very much for the review of the manuscript "Sustainable Developmnet Planning: Master's Based on Project-Based Learning Approach".

All your comments have been incorporated and have been very useful to significantly improve the paper.

The strong improvements made are listed below:

1) The introduction has been improved focusing more on the objective of the paper, explaining the novelty of the research.

We specify that this is a case study, which includes finding only based on one case study, Erasmus Mundus Master (International Master of Sustainable Development), in a highly international context.

2) The philosophical-historical background (descriptive research) has been greatly reduced, focusing on the objectives of the research.

3) The research methodology has been clarified, rewritten and abbreviated. The research methodology has been described better and more clearly

4) The research objectives and the research hypothesis have been specified.

5) The research objectives and the research hypothesis have been recapitulated in the results, implications and conclusions.

6) Details on the case study have been simplified by focusing on the key aspects to repeat the research to other situations. Conceptual schemes have been developed to help understand the experience, generalize and replicate the experience to other environment (other masters, universities, and countries)

7) The results' chapter has been greatly shortened and rewritten, mixes up results, theoretical framework, and information about the program. The results are presented in a clearer and more objective way through conceptual schemes on the findings. For clarity, a short section on numerical results has been created, and another section on the implications of the results. More interpretations of the results are provided. More comments on the results and comparisons with similar literature studies have been included.

8) The literature review on other experiences has been extended, especially those from journals indexed in international databases. The research objectives and the research hypothesis have been recapitulated in the conclusions. A deep and profound rewriting of the article has been made.

Thank you very much for your contributions that have been fundamental for the improvement of the quality of the article.

Reviewer 2 Report

Review of "Sustainable Developmnet Planning: Master's Based on Project-Based Learnign Approach"

My evaluation of "Sustainable Developmnet Planning: Master's Based on Project-Based Learnign Approach" will actually be an evaluation of several of its parts, because I rate these parts quite differently. Let's focus on the individual parts and try to identify their strengths and weaknesses, and this evaluation will lead to an overall evaluation of the article as a whole.

In the section “1. Introduction” is presented main idea of ​​the article. Unfortunately, this part is quite proclamative and not at all very specific in most parts. I would recommend here much more specific identification of concrete objectives that the article wants to achieve, and to show stronger (more general claims) that they are based on more than just proclaimable claims.

I consider as the most problematic part “2. Conceptual Background '. In the beginning, this part seeks a certain philosophical-historical background, the theoretical anchoring of the subsequent empirical part of the work. Why do I consider it problematic and propose its fundamental modification? 1. It tries to solve too many problems at once. In fact, it is not aimed at creating and underpinning theoretical bases for the empirical part of the thesis itself, but it addresses issues such as: modernity and postmodernity, the philosophical foundations of modernity, both on extremely small (and inadequate) space; in other parts it is quite practical, and very often without the interconnection of this two approaches, and also without the connection that would create a logical interspace. Conceptually conceive the difference between modern, postmodern and contemporary, to link the methodological approaches of the text to these concepts to indicate where the limits of modernity are (by the way the concept of modernity as presented by the authors is not the most used one within sociology, social psychology or philosophy). Simply speaking it is not possible to solve such a “big questions” using just two and half pages so text logically ends in a lot of rather proclamatory and simplistic statements. This critical evaluation mainly concerns Part 2.1, but unfortunately 2.2 and 2.3 are only slightly better. In addition, all three parts contain a large number of "positive" words and floscules and little real theoretical-preparatory work for the empirical part itself.
However, the quality of the text will significantly change in Section 3, where the emphasis of the article begins to describe and later analyze a particular university degree program and the impact of its concept and structure on acquiring the specific competencies and knowledge of students who have completed it. This part is disproportionately better both its anchoring and, above all, the quality of the structure and the specificity of the provided information changes the reader's impression - suddenly we read a text that refers to specific sources of information, brings further knowledge and is able to work well with the obtained empirical material. Only in a few passages does the quality of genre change, which in my opinion refers to the use of texts from projects created at the start of the study program. Therefore, I recommend removing or rewriting these passages. In addition, in terms of the objectives of the text, I do not fully understand the benefits of Table 1, in which we monitor the number of enrolled and enrolled students with another criterion (internationality).
The methodological part of the thesis is also significantly better than the introduction, but it seems to me that it certainly deserves further reading and little bit rewritting (more anchorage in reference to the adopted methods) and abbreviation.
The empirical findings of the study themselves are interesting, bringing both concrete knowledge and, in some places, a relatively successful effort for an analytical and synthetic elaboration of these particular findings, I miss (and I would definitely recommend) further comparisons with findings in similarly focused scientific texts. Even (but this is pure speculation) I have the impression that the program itself is really well prepared and has such a historical record that it had a positive impact on the quality of these passages.
The text is basically a case study with some theoretical overlap, and a part of the case study itself is really acceptable, bringing new knowledge and very well structured on a large scale, ending with an analytical and synthetic extension. 

However, in the introductory, theoretical and research part of the work, the study has in my opinion really serious shortcomings, whether it is an effort to solve too general (and to the topic only loosely related) questions, too simplistic statements and arguments. So here I recommend deep and profound rewriting.

Author Response

Dear reviewer,

Thank you very much for the review of the manuscript "Sustainable Developmnet Planning: Master's Based on Project-Based Learning Approach".

All your comments have been incorporated and have been very useful to significantly improve the paper.

The strong improvements made are listed below:

1) The introduction has been improved focusing more on the objective of the paper, explaining the novelty of the research. We specify that this is a case study, which includes finding only based on one case study, Erasmus Mundus Master (International Master of Sustainable Development), in a highly international context.

2) According to your suggestions, the most problematic part (the philosophical-historical background, has been greatly reduced, focusing on the objectives of the research.

3) The research methodology has been clarified, rewritten and abbreviated. The research methodology has been described better and more clearly. The research objectives and the research hypothesis have been specified. The research objectives and the research hypothesis have been recapitulated in the results, implications and conclusions.

4) Details on the case study have been simplified by focusing on the key aspects to repeat the research to other situations.

5) Conceptual schemes have been developed to help understand the experience, generalize and replicate the experience to other environment (other masters, universities, and countries)

6) The empirical findings / results' chapter has been greatly shortened and rewritten, mixes up results, theoretical framework, and information about the program. The results are presented in a clearer and more objective way through conceptual schemes on the findings. For clarity, a short section on numerical results has been created, and another section on the implications of the results. More interpretations of the results are provided. More comments on the results and comparisons with similar literature studies have been included.

7) The literature review on other experiences has been extended, especially those from journals indexed in international databases. The research objectives and the research hypothesis have been recapitulated in the conclusions. A deep and profound rewriting of the article has been made.

Thank you very much for your contributions that have been fundamental for the improvement of the quality of the article.

Reviewer 3 Report

General comment: This paper provided an analysis of the teaching strategies.

Introduction: The Introduction should be improve focusing on the aim of the paper, main methods, main results and few recommendations based on empirical results. The authors should explain their research novelty compared to previous studies from literature.
Methodology: Indicate some quantitative methods useful for this study, indicate limits and advantages of methods. Indicate alternative methods. Provide practical comments to introduce the methods.
Results: The interpretations are too superficial and are not based on quantitative method. More comments of the results are required and comparisons with similar studies from literature. More details on data are required.
Discussion: Interpretations of the results are provided, but a more critical position is required. The literature review should be extended.
Bibliography/References: The reference list is not up-to-date. Add recent references, especially those from journals indexed in international databases, WoS and Scopus.

Author Response

Dear reviewer,

Thank you very much for the review of the manuscript "Sustainable Developmnet Planning: Master's Based on Project-Based Learning Approach".

All your comments have been incorporated and have been very useful to significantly improve the paper.

The strong improvements made are listed below: 

1) Introduction.- 

The introduction has been improved focusing more on the objective of the paper, explaining the novelty of the research. We specify that this is a case study, which includes finding only based on one case study, Erasmus Mundus Master (International Master of Sustainable Development), in a highly international context. The philosophical-historical background, has been greatly reduced, focusing on the objectives of the research.

2) Methodology

The research methodology has been clarified, rewritten and abbreviated. The research methodology has been described better and more clearly. The research objectives and the research hypothesis have been specified. The research objectives and the research hypothesis have been recapitulated in the results, implications and conclusions.

3) Results and Discussion

The results are presented in a clearer and more objective way through conceptual schemes on the findings. For clarity, a short section on numerical results has been created, and another section on the implications of the results. The results' chapter has been greatly shortened and rewritten, mixes up results, theoretical framework, and information about the program. Conceptual schemes have been developed to help understand the experience, generalize and replicate the experience to other environment (other masters, universities, and countries). More interpretations of the results are provided. More comments on the results and comparisons with similar literature studies have been included.

4) Bibliography/References/Conclusions.

The literature review on other experiences has been extended, especially those from journals indexed in international databases. The research objectives and the research hypothesis have been recapitulated in the conclusions.

A deep and profound rewriting of the article has been made

Thank you very much for your contributions that have been fundamental for the improvement of the quality of the article.

Round 2

Reviewer 1 Report

The article has been significantly improved both methodologically and theoretically.

The 2 figures look not correct and with stage characters. They need to be reviewed.

Author Response

Dear reviewer,

Thank you very much for the new revision of the manuscript.

The 2 figure has been reviewed and corrected

Thank you very much for your contributions that have been fundamental for the improvement of the quality of the article.

Best regards!

Reviewer 3 Report

Accept after minor revision

The reference list is not up-to-date. Add recent references, especially those from journals indexed in international databases, WoS and Scopus. 

Author Response

Dear reviewer,

Thank you very much for the new revision of the manuscript.

The list of references has been updated and expanded with recent references: have been expanded with 12 new experiences, from journals indexed in international databases, (JCR, WoS and Scopus). The experiences are particularly about the implementing Sustainable Development in a Higher Education context and in the world-class university context.

Thank you very much for your contributions that have been fundamental for the improvement of the quality of the article.
